# Continuous Radiofrequency for Morton’s Neuroma: Is There Complete Ablation? A Preliminary Report

**DOI:** 10.3390/healthcare13151838

**Published:** 2025-07-28

**Authors:** Gabriel Camuñas-Nieves, Alejandro Fernández-Gibello, Simone Moroni, Felice Galluccio, Mario Fajardo-Pérez, Francisco Martínez-Pérez, Eduardo Simón-Pérez, Alfonso Martínez-Nova

**Affiliations:** 1Clínica Vitruvio, C/María de Guzmán 47, 28003 Madrid, Spain; gabrielcamunas@gmail.com (G.C.-N.); alejandrofernandezgibello@gmail.com (A.F.-G.); fran.martinezperez@gmail.com (F.M.-P.); 2Podiatry Department, San Vicente Mártir University, Pl. Almoina 3, 46003 Valencia, Spain; dott.simonemoroni@gmail.com; 3Rheumatology and Pain Management, Fisiotech Lab Studio, 50136 Firenze, Italy; felicegalluccio@gmail.com; 4Center for Regional Anesthesia and Pain Medicine (CRAPM), Wan Fang Hospital, Taipei Medical University, Taipei 116, Taiwan; 5Ultradissection Group, 28003 Madrid, Spain; mfajardoperez@yahoo.es; 6Centro Médico Recoletas, C/Angustias 19, 47003 Valladolid, Spain; eduardo.simon.gmp@gruporecoletas.com; 7Department of Nursing, University of Extremadura, 10600 Plasencia, Spain

**Keywords:** neuroma, radiofrequency, nerve ablation, Morton’s neuroma, histological analysis

## Abstract

**Background and Objectives:** Morton’s neuroma is a painful foot condition that can be treated with continuous radiofrequency. However, its efficacy is not always optimal, with failure rates of 15–20%. It has been suggested that these failures may be due to incomplete nerve ablation, allowing for nerve regeneration and persistent pain. So, the aim of this study was to assess the histological effects of continuous radiofrequency on the nerves affected by Morton’s neuroma. **Materials and Methods:** The effect of continuous radiofrequency was evaluated in two patients with Morton’s neuroma, which required open surgery excision. In both cases, radiofrequency with a standard protocol was applied ex vivo, following the surgical excision of the neuroma. A TLG10 RF generator (90 °C, 90 s) with a monopolar needle with a 0.5 cm active tip was used. Subsequently, the samples were histologically analyzed to determine the degree of nerve ablation. **Results:** Histological analysis showed homogeneous focal necrosis in both cases, with lesion depths of 2.4 mm and 3.18 mm. However, areas of intact nerve tissue were identified at the periphery of the neuroma, suggesting incomplete ablation. **Conclusions:** The findings indicate that continuous radiofrequency does not guarantee total nerve ablation, which could explain recurrence in some cases. Intraoperative neurophysiological monitoring could be key to optimizing the procedure, ensuring complete interruption of nerve conduction and improving treatment efficacy.

## 1. Introduction

Morton’s neuroma is a painful condition affecting the common plantar digital nerves of the foot, often associated with tight footwear and repetitive microtrauma [1]. It is characterized by perineural thickening that causes radiating pain towards the toes or metatarsal pain [2,3]. Conventional treatments include orthopedic insoles, corticosteroid injections, and changes in footwear. However, their effectiveness decreases depending on the size of the neuroma and the duration of symptoms [4,5]. When conservative options fail, surgical alternatives or minimally invasive procedures like radiofrequency are available.

Traditional surgical approaches have variable success rates and may involve complications, such as fibrosis of the tissues, stump neuroma, instability in the metatarsophalangeal joints, hypoesthesia, or dysesthesia [6,7]. Techniques such as ultrasound-guided or endoscopic surgery have shown better results in terms of precision and recovery [8,9]. Nevertheless, continuous radiofrequency remains a highly effective alternative, with success rates ranging from 80% to 85%, offering an effective option with more favorable recovery for neuroma treatment [10,11]. The mechanism of action of radiofrequency is based on the generation of heat through radio waves (electromagnetic-induced current) emitted from a needle with an active tip of 0.5 or 1 cm. Controlled heat is produced by the agitation and friction of ions in the tissues near the needle. The result is the denaturation of nerve tissue proteins, causing coagulation and tissue destruction. This is known as thermal ablation and interrupts nerve stimuli, preventing the nerve from sending pain signals [12,13].

The failure rate of the radiofrequency technique varies between 15% and 20%. This failure rate could be due to incomplete nerve ablation, as complete denervation should theoretically result in no sensitivity in the innervated area, as seen in rats [14]. Currently, the protocol involves treatment at 90 °C for 90 s. We believe that nerve ablation might not be complete in this percentage of failed cases, where ablation is not applied throughout the neural and perineural extent. This failure rate could be influenced by the application area of the radiofrequency technique, being applied directly on nerve fibrosis instead of on the healthy nerve in a more proximal area. Additionally, some variations in the current protocol might improve the results, such as applying the radiofrequency technique in different areas to increase the exposure area, as well as the number of applications. Maybe more applications increase the ablation area, thus improving the chances of success.

Although continuous radiofrequency aims for complete nerve ablation to alleviate symptoms, there is no histopathological evidence that all nerve tissues have been burned. This suggests that in 15–20% of failed cases, radiofrequency may not have been able to burn the entire nerve. This limits the understanding of its mechanism of action. Therefore, the objective of this study is to evaluate the ablative effect of continuous radiofrequency on Morton’s neuroma and analyze whether its size affects its effectiveness.

## 2. Case Reports

This study was approved by the Research Ethics Committee of the University of Extremadura in accordance with the principles outlined in the Declaration of Helsinki (ID 97//2022). The sample consisted of two patients who, due to their clinical characteristics, required neuroma excision. Both patients had undergone unsuccessful conservative treatment for at least two years prior to surgical intervention. Written informed consent was obtained from each participant for both the surgical procedure and their inclusion in the study. Preoperative ultrasonographic measurements of the neuroma were performed to document its size prior to excision. Imaging was conducted in both short- and long-axis views, focusing on the interdigital space between the third and fourth metatarsal heads of the right foot. A GE LOGIQ R7 (General Electric Healthcare, Lynn, MA, USA) ultrasound system equipped with a 12 MHz linear transducer was used for the assessments.

### 2.1. Case 1

A 44-year-old female patient at the time of surgery presented with bilateral Morton’s neuromas. The neuroma in the left foot was asymptomatic, while the neuroma in the right foot was associated with recurrent pain over the previous five years. No relevant medical history or known drug allergies were reported. Based on short-axis ultrasound imaging, the neuroma appeared as a hypoechoic mass measuring 1.18 mm in width (Figure 1, Top), and it was visualized during Mulder’s maneuver. Over the last five years, the pain had progressed from mild discomfort to a completely disabling condition during the final year, despite conservative management including orthotic interventions and corticosteroid infiltrations with betamethasone. Surgical excision of the neuroma was ultimately indicated.

### 2.2. Case 2

A 52-year-old female patient presented with a Morton’s neuroma in the right foot, associated with a three-year history of painful symptoms. Multiple attempts at conservative treatment—including footwear modifications and custom insoles—failed to provide symptom relief. Although the patient experienced a slight improvement with orthotic insoles, the outcome was ultimately unsatisfactory. In this second case, the neuroma was located in the third intermetatarsal space and measured 0.74 mm in width according to short-axis ultrasonography (Figure 1, Bottom). Mulder’s sign was also positive.

### 2.3. Surgical Technique

Surgical excision of the neuroma via a dorsal approach was performed. In both cases, a longitudinal dorsal approach was performed over the affected intermetatarsal space, approximately 3 cm in length, following skin tension lines to minimize scarring (Figure 2). Layered dissection was carried out, carefully identifying and preserving the surrounding neurovascular structures. The transverse intermetatarsal ligament was exposed and meticulously sectioned to access the neuroma. Once the hypertrophied common digital plantar nerve was identified, proximal dissection was performed until healthy nerve tissue was reached. The nerve was then transected using a scalpel, and the proximal stump was secured. Electrocautery was subsequently applied to minimize the risk of symptomatic stump neuroma formation.

Meticulous hemostasis was achieved, and layered wound closure was performed. The deep fascia and subcutaneous tissue were approximated using simple interrupted absorbable sutures (polyglactin 910, Vicryl 4-0). The skin was closed with an absorbable intradermal suture to optimize the esthetic outcome and reduce the risk of wound dehiscence. Finally, a moderate compressive dressing was applied to ensure tissue stability and minimize postoperative inflammation.

### 2.4. Application of Radiofrequency

Following resection, ex vivo continuous radiofrequency ablation was applied to the excised neuroma. Using a monopolar needle with a 0.5 cm active tip, radiofrequency ablation was performed at a temperature of 90 °C for 90 s, employing the TLG10 RF Generator (manufactured by Top, Medikey, Gouda, The Netherlands, 2021). The needle was inserted longitudinally along the axis of the neuroma (Figure 3), with the tip positioned at its central portion. The depth of insertion was adjusted so that the active tip was fully housed within the nerve tissue. Care was taken to maintain a parallel orientation to the fascicular structure in order to optimize thermal propagation along the nerve axis. These parameters were kept constant in both cases to replicate, as closely as possible, standard in vivo clinical conditions.

Upon completion of the procedure (Figure 3), the specimens were submitted to the Department of Pathology for histological evaluation of radiofrequency-induced thermal damage. Special attention was given to the depth of the thermal lesion and the extent of axonal degeneration. The objective of this analysis was to determine whether the neuroma’s size influenced the effectiveness of the radiofrequency treatment, considering the relationship between neuroma dimensions and the ablation diameter achieved by the needle.

## 3. Results

In the first case, the specimens obtained after ex vivo radiofrequency ablation exhibited homogeneous thermal necrosis, with a maximum depth of 2.4 mm. Hematoxylin–eosin staining revealed a characteristic pattern of coagulative necrosis, with extensive destruction of neural tissue in areas subjected to the greatest thermal exposure. In the corresponding histological image (Figure 4), the darker regions represent areas where radiofrequency caused the most significant thermal damage, resulting in substantial ablation of nervous tissue. These areas demonstrated marked structural loss, collagen denaturation, and disruption of the organization of nerve fascicles.

In contrast, the lighter regions indicate areas where ablation was incomplete, preserving the structural integrity of some neural components. This suggests that although radiofrequency ablation induced homogeneous necrosis in the central portion of the neuroma, the effect decreased toward the periphery, allowing the survival of some peripheral nerve fascicles. In the second case, which also involved ex vivo radiofrequency ablation, more extensive thermal necrosis was observed, reaching a depth of up to 3.18 mm. Despite the increased depth of tissue injury, the histological pattern maintained a homogeneous distribution of necrosis, with uniform involvement across the ablation zone. In the corresponding histological image (Figure 4), the darker regions once again reflect areas exposed to higher temperatures and thus greater thermal damage. These areas demonstrated deeper and more extensive necrosis, with structural collapse of neural tissue and complete loss of cellular nuclei. Unlike the first case, a reduced amount of viable neural tissue was observed at the periphery, suggesting that in this patient, ablation reached a larger proportion of the neuroma. Nevertheless, some lighter peripheral zones were still present, indicating that despite the greater extent of necrosis, complete ablation of the neuroma was not achieved.

## 4. Discussion

Continuous radiofrequency (RF) ablation for the treatment of Morton’s neuroma has demonstrated a relatively high success rate in pain reduction, with previous studies reporting pain relief rates between 80% and 85% [10,11]. Also, the production of necrosis using radiofrequency has been demonstrated in rat nerves [14].

However, the ability of RF to achieve complete neuroma ablation may be limited by the size of the lesion. Only one study to date has provided evidence that neuroma size could be a limiting factor in the effectiveness of various treatments [15]. The maximum depth of necrosis generated by RF is approximately 3 mm [13,16], which may be insufficient to completely ablate larger neuromas. This could explain the suboptimal outcomes in 15–20% of cases, or why some studies have shown that multiple applications at different points are more effective [17].

Additionally, the use of a needle with a 1 cm active tip, as opposed to the 0.5 cm tip used in this study, may improve results by increasing the ablation area. Another important factor is the location of the lesion. Techniques using fluoroscopic guidance have shown similar results to those using ultrasound guidance. However, fluoroscopy-guided RF is typically performed more proximally, which may result in damage to healthy nerve tissue. In contrast, ultrasound-guided RF allows for a more distal approach, targeting the neuroma itself while avoiding damage to unaffected nerve segments, potentially increasing the precision and safety of the procedure.

Despite identical RF settings, we found different ablation sizes in the two specimens. The smaller neuroma in Case 2 resulted in a larger ablation area compared to the larger neuroma in Case 1. The discrepancy in thermal lesion size may be due to the histological sectioning plane. An oblique cut in Case 2 may have intersected a wider area of the necrotic zone, making the lesion appear larger, despite identical radiofrequency settings in both cases. Also, it could also be hypothesized that larger neuromas contain a greater proportion of fibrotic tissue, which may hinder the effective induction of thermal necrosis.

An improvement to the technique was already proposed in the study “Three RF applications are better than two”, where applying RF at different positions resulted in better clinical outcomes. This supports the theory that the lesion area created by RF is limited to a diameter of approximately 3 mm. This concept is crucial, as performing RF at a single site or simply increasing the duration of ablation does not expand the area of effect [17]. However, increasing the temperature above 90 °C in an attempt to enlarge the lesion could be counterproductive, as it may induce cell death in healthy tissue and exacerbate inflammation [13].

Intraoperative neurophysiological monitoring may be a key tool for optimizing RF treatment outcomes. This technique allows the clinician to adjust RF parameters in real time, such as the number and positioning of applications, thereby increasing the likelihood of achieving complete neural ablation. Although neurophysiological monitoring is typically used to preserve nerve function in surgery, its purpose in this context is the opposite: to ensure that the nerve has been entirely disrupted and no longer conducts impulses. A real-time assessment of the neural response enables confirmation of complete signal interruption, minimizing the risk of nerve regeneration and the recurrence of pain [18]. Thus, intraoperative monitoring could enhance the efficacy of RF by ensuring a definitive treatment.

In summary, although continuous RF is effective in reducing pain associated with Morton’s neuroma, it does not guarantee complete ablation of the neural tissue. Technique modifications, such as using needles with longer active tips, targeting more proximal segments of the neuroma, and performing multiple ablations at different sites, may improve outcomes. Furthermore, intraoperative neurophysiological monitoring offers an additional layer of control during the procedure, enabling more precise and reliable ablation.

Although our study focuses on the histological effects of radiofrequency, it is important to consider that some cases of clinical failure may result from initial diagnostic inaccuracies, particularly in patients with nonspecific forefoot symptoms. In addition, repeated radiofrequency procedures may cause structural disruption of the perineurium, potentially leading to secondary neuroma formation [19]. These factors cannot be assessed in an ex vivo model but should be taken into account in clinical practice.

This study has some limitations, and the results must be interpreted with caution. The procedure was performed under direct visualization. However this procedure mimics in vivo ultrasound guidance. But this ex vivo model replicates real-world application. Also, another limitation of ex vivo is the inability to assess functional nerve loss. 

We suggest future studies involving histopathological analysis of neuromas excised after failed in vivo RF treatment, which could provide more clinically relevant insights. Also, further studies should investigate whether there is a specific neuroma size threshold beyond which RF treatment becomes less effective.

## 5. Conclusions

In both cases, ex vivo radiofrequency ablation resulted in homogeneous thermal necrosis, with differences in the depth of injury observed. This study supports the notion that continuous radiofrequency is an effective tool in the management of Morton’s neuroma, though it presents limitations, particularly in larger neuromas. The histological images clearly show that the darker areas correspond to regions exposed to higher temperatures and therefore exhibit greater thermal damage, while the lighter areas reflect residual neural tissue that was not completely ablated. Despite the overall homogeneous distribution of injury in both specimens, the presence of intact tissue at the neuroma’s margins indicates that radiofrequency ablation, although effective in the central portion, does not consistently reach the entirety of the neural structure. The results suggest that neuroma size and the RF application technique play a crucial role in treatment effectiveness. Likewise, greater proximity of the needle to the nerve and multiple ablations at different points along the healthy nerve segment may yield more satisfactory outcomes.

## Figures and Tables

**Figure 1 healthcare-13-01838-f001:**
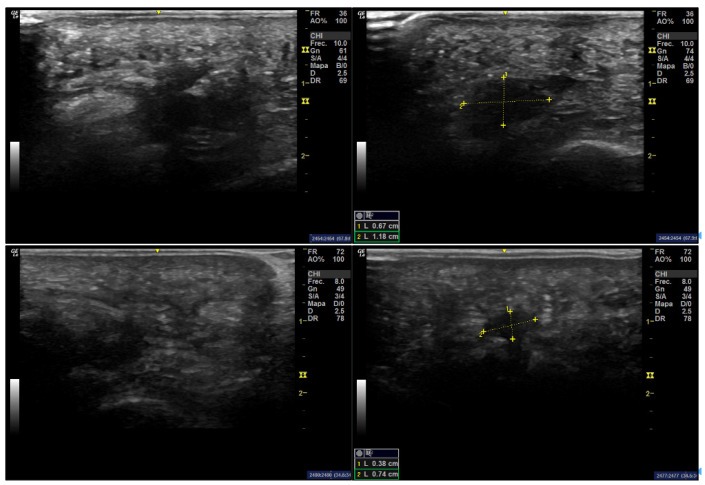
Ultrasound scans of a Morton’s neuroma in the third intermetatarsal space. Top: Case 1. Longitudinal (left) and transverse (right) ultrasound views showing a neuroma measuring 1.18 × 0.67 cm. Bottom: Case 2. Longitudinal (left) and transverse (right) ultrasound views showing a neuroma measuring 0.74 × 0.38 cm. In both cases, a well-defined hypoechoic structure is visualized between the metatarsal heads.

**Figure 2 healthcare-13-01838-f002:**
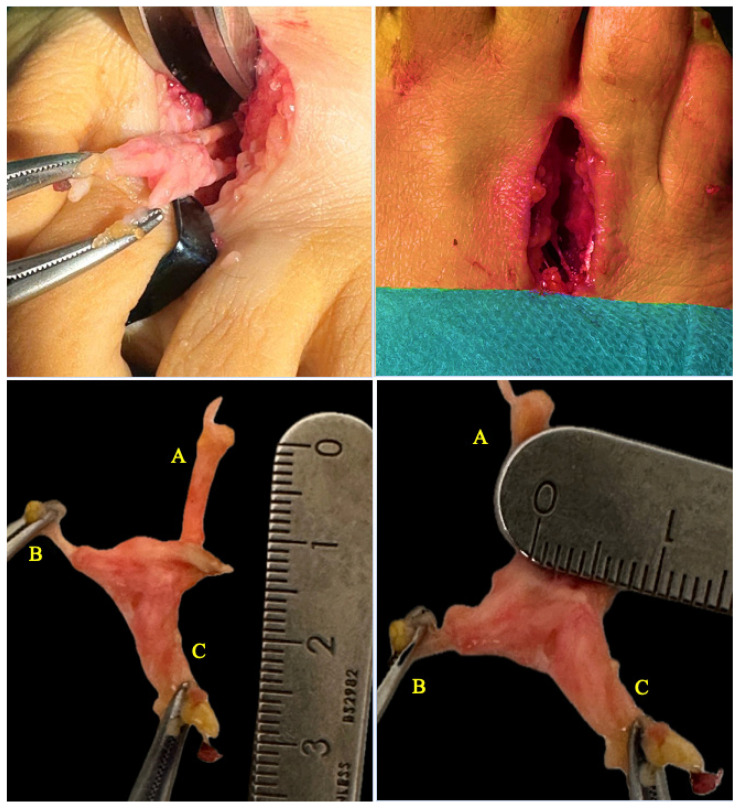
Intraoperative and anatomical images corresponding to a Morton’s neuroma in the third intermetatarsal space following surgical excision (Case 1). Top: Surgical exposure of the interdigital nerve via a dorsal approach. Bottom: Macroscopic view of the neuroma with three identifiable nerve branches. Three main nerve branches are identified: A: third interdigital branch. B: medial plantar digital. C: third intermetatarsal common nerve.

**Figure 3 healthcare-13-01838-f003:**
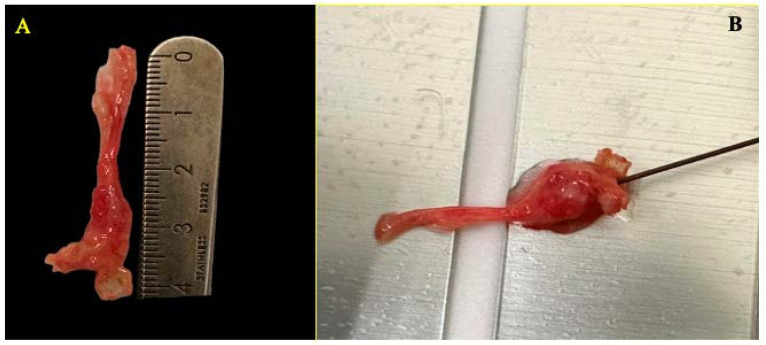
Ex vivo continuous radiofrequency procedure on a Morton’s neuroma. (**A**) Macroscopic image of the specimen after surgical excision, with longitudinal measurement of the neuroma using a millimeter scale. (**B**) Radiofrequency application using an active needle positioned directly on the neuroma. The metallic background corresponds to the return plate of the generator, which was required to complete the electrical circuit and enable thermal transmission during the ablation process.

**Figure 4 healthcare-13-01838-f004:**
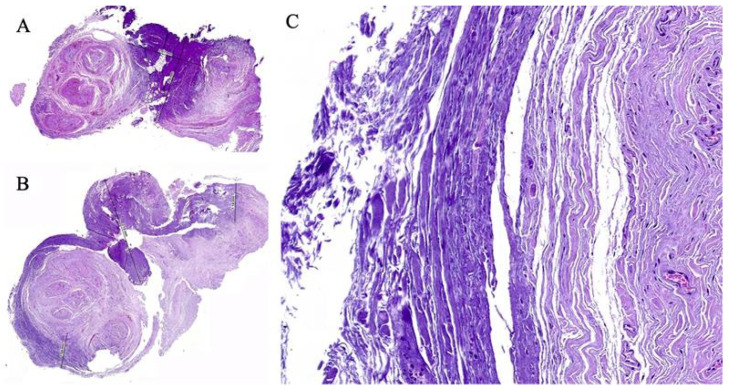
Histological sections of a Morton’s neuroma following radiofrequency application, stained with hematoxylin and eosin. The darker purple regions represent areas where radiofrequency caused the most significant thermal damage, resulting in substantial ablation of nervous tissue. (**A**) Panoramic section from Case 1 showing partial thermal necrosis up to 2.4 mm in depth. (**B**) Panoramic section from Case 2 demonstrating more extensive necrosis reaching 3.18 mm, with a reduced amount of viable neural tissue. (**C**) Magnified image from Case 1 showing disorganized nerve fascicles, perineural fibrosis, and tissue degeneration consistent with thermally induced damage.

## Data Availability

Data is contained within the article.

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
