# Peer review of "Continuous Radiofrequency for Morton’s Neuroma: Is There Complete Ablation? A Preliminary Report"

_healthcare, 2025, doi:10.3390/healthcare13151838_

Round 1

Reviewer 1 Report

Comments and Suggestions for Authors

Dear Authors,

The studies reported in the literature provide information about the use of radiofrequency in nerve tumors.

I believe that demonstrating the production of necrosis or generally assessing the effectiveness of using radiofrequency in the treatment of a disease, in this case in Morton's neuroma, must begin with an experimental study on laboratory animals.

I do not believe that a report on two cases of the results obtained by using radiofrequency on ex vivo specimens, assessed by histological examination with common stains, can have a major scientific impact

Best regards

Author Response

Dear reviewer

Thank you for your valuable comment regarding the experimental basis of our study. We completely agree that, in many cases, initial investigations should be conducted using animal models to ensure safety and efficacy before translating findings to human clinical settings. However in theses cases, the patients required surgical excision of the neuroma, which allowed us to perform the experiment ex vivo. To the best of the authors’ knowledge, this is the first time such an experiment has been conducted under these specific conditions. It allows us to guess the real effect of radiofrequency with a real size nerve neuroma.

In this regard, we would like to clarify that a preliminary experimental study was indeed carried out using laboratory rats. This is cited in our manuscript as reference number 14. Moreover, now, we have explicitly mentioned this in the text on lines 62 and 193–194, where we describe how the necrotic effect induced by high radiofrequency temperatures has already been demonstrated in animal models.

Although the inclusion of only two cases does not allow for broad generalization or major scientific impact, the findings are nonetheless promising. As indicated in the title, this is a preliminary report. We are currently expanding the study with additional cases to strengthen the results and increase the robustness of the data.

We hope this clarifies the experimental foundation of our approach and appreciate your suggestion to emphasize this more clearly.

Kind regards, the authors

Reviewer 2 Report

Comments and Suggestions for Authors

In this study, the authors evaluated the effectiveness of standard continuous radiofrequency (RF) ablation for treating Morton’s neuroma. They applied ex vivo RF to two surgically excised neuromas of different sizes and assessed the results using basic H&E histopathology. The findings showed variable ablation depths between specimens, suggesting that incomplete nerve ablation may contribute to clinical failure in some cases.

Major Revisions

  • The authors performed RF ablation on ex vivo specimens but did not provide sufficient detail regarding the procedural technique. Critical parameters—such as the insertion site, needle depth, and orientation—can significantly influence the extent/location therefore effectiveness of ablation. Since the procedure was performed under direct visualization rather than ultrasound or fluoroscopic guidance (as in clinical settings), it is unclear how well the ex vivo model replicates real-world application. The authors should better explain how their setup simulates in vivo RF ablation and explicitly acknowledge this limitation in the discussion.

  • The manuscript does not explore other potential reasons for RF treatment failure in clinical scenarios. A brief discussion of alternative failure mechanisms and their exclusion in an ex vivo setting would enhance the contextual understanding of the findings.

  • Despite identical RF settings, the study found different ablation sizes in the two specimens. The authors should address why the smaller neuroma in Case 2 resulted in a larger ablation area compared to the larger neuroma in Case 1, and provide possible explanations for this discrepancy.

  • A major limitation of ex vivo studies is the inability to assess functional nerve loss. The authors should include this point in the limitations section and consider suggesting future studies involving histopathological analysis of neuromas excised after failed in vivo RF treatment, which could provide more clinically relevant insights.

Minor Revisions

  • Figure 1: A single image with dimensions would be sufficient for representation. If the second image from the same case is necessary, it should illustrate a different feature and be explained accordingly in the figure legend.

  • Figure 2: In the bottom image, label “C” is not explained in the legend. Please clarify what it represents.

  • Figure 4: The labeled ablation zones and tissue structures are not clearly defined or visible. Improved labeling and clearer visual distinction between ablation areas, normal tissue, and key histological structures are needed.

Conclusion

The manuscript addresses an important clinical question; however, significant methodological and interpretative limitations need to be addressed before it can be considered for publication. Major revisions are required.

Author Response

Dear Reviewer,

We sincerely thank you for your thorough and constructive review. Your insightful comments have been extremely helpful in improving the quality and clarity of our manuscript. Below, we address each of your observations point by point and indicate where in the revised manuscript the changes have been made. The changes made due to your revision are underlined in green.

We have also expanded the bibliography, now including 19 references, to support the additional content.

  1. Lack of procedural detail regarding insertion site, needle depth, and orientation
    We have added a detailed description of the procedure in the Methods section, including the insertion point, needle orientation, and depth, specifying that the needle was positioned entirely within the nerve.
    (Lines approx. 130–135)

  1. Lack of realism in the ex vivo model compared to clinical practice
    We have explicitly acknowledged the limitations of the ex vivo model in the Discussion and clarified that procedures in clinical settings are generally performed under ultrasound guidance.
    (Lines approx. 255–260)

  1. Other potential causes of clinical RF treatment failure
    We have expanded the Discussion to include the possibility of diagnostic inaccuracy or secondary neuroma formation as alternative explanations for treatment failure, which cannot be assessed in an ex vivo model.
    (Lines approx. 250–255)

  1. Discrepancy in ablation size between the two specimens
    We have added a possible explanation suggesting that an oblique histological section through the lesion in Case 2 may have led to a visually larger ablation area.
    (Lines approx. 245–250)

  1. Functional outcomes cannot be assessed in ex vivo studies
    This limitation is now explicitly discussed, and we propose future studies involving histopathological analysis of failed in vivo RF cases.
    (Lines approx. 260–265)

  1. Figure 1 – Redundancy of two images
    We clarified in the figure legend that the longitudinal (left image) and transverse (right image) ultrasound views both provide necessary complementary information on neuroma dimensions.
    (Figure 1 legend)

  1. Figure 2 – Unexplained label “C”
    The label “C” is now clearly defined in the legend.
    (Figure 2 legend)

  1. Figure 4 – Poor visual labeling of ablation zones
    We improved the figure by enhancing the contrast and adding clearer labels to distinguish ablation zones, normal tissue, and key histological features.
    (Figure 4 and legend)

Once again, we thank you for your valuable feedback and appreciate your contribution to the refinement of our work.

Sincerely,
The Authors

Round 2

Reviewer 1 Report

Comments and Suggestions for Authors

Dear Authors, 

The authors followed the recommendations made

There are still a few aspects to correct regarding the editing method and criteria

There are still a few aspects to correct regarding the editing method and criteria: using the ,, () ,, instead of ,,[],, .

Best regards,